# RE-IMAGINE THE NEGATIVE PROMPT ALGORITHM FOR 2D/3D DIFFUSION

## ABSTRACT

Although text-to-image diffusion models have made significant strides in generating images from text, they are sometimes more inclined to generate images like the data on which the model was trained rather than the provided text. This limitation has hindered their usage in both 2D and 3D applications. To address this problem, we explored the use of negative prompts but found that the current implementation fails to produce desired results, particularly when there is an overlap between the main and negative prompts. To overcome this issue, we propose Perp-Neg, a new algorithm that leverages the geometrical properties of the score space to address the shortcomings of the current negative prompts algorithm. Perp-Neg does not require any training or fine-tuning of the model. Moreover, we experimentally demonstrate that Perp-Neg provides greater flexibility in generating images by enabling users to edit out unwanted concepts from the initially generated images in 2D cases. Furthermore, to extend the application of Perp-Neg to 3D, we integrate Perp-Neg with the state-of-the-art text-to-3D (DreamFusion) method. Our experimental studies clearly show the effectiveness of Perp-Neg in addressing the Janus (multi-head) problem. Perp-Neg has enabled the generation of 3D assets that were previously unattainable due to the persistent Janus problem, even after multiple attempts.

## 1 INTRODUCTION

Advancements in generating images using diffusion models from text have shown remarkable capabilities in producing a wide range of creative images from unstructured text inputs Balaji et al. (2022); Ramesh et al. (2022); Rombach et al. (2022); Saharia et al. (2022); Yu et al. (2022). However, research has found that the generated images may not always accurately represent the intended meaning of the original text prompt Brooks et al. (2022); Chefer et al. (2023); Hertz et al. (2022); Wang et al. (2022b).

Generating satisfactory images that semantically match the text query is challenging, as it requires textual concepts to match the images at a grounded level. However, due to the difficulty of obtaining such a fine-grained annotation, current text-to-image models have difficulty fully understanding the relationship between text and images. Therefore, they are inclined to generate images like high-frequent text-image pairs in the datasets, where we can observe that the generated images are missing requested or containing undesired attributes Li et al. (2023). Most of the recent works focus on adding back the missing objects or attributes to existing content to edit images based on a well-designed main text prompt Alt et al. (2022); Brooks et al. (2022); Chefer et al. (2023); Couairon et al. (2022); Gal et al. (2022); Kawar et al. (2022); Lugmayr et al. (2022); Meng et al. (2021); Su et al. (2022). However, limited of them study how to remove redundant attributes, or force the model *NOT* to have an unwanted object using negative prompts Du et al. (2020), which is the main goal of our paper.

We start this paper by showing the shortcomings of the current negative prompt algorithm. After our initial investigation, we realized the current implementation of the negative prompt could produce unsatisfactory results when there is an overlap between the main prompt and the negative ones, as shown in the examples in Figure 1. To address the above problem, we propose Perp-Neg algorithm, which does not require any training and can readily be applied to a pre-trained diffusion model. We refer to our method as Perp-Neg since it employs the perpendicular score estimated by the denoiser

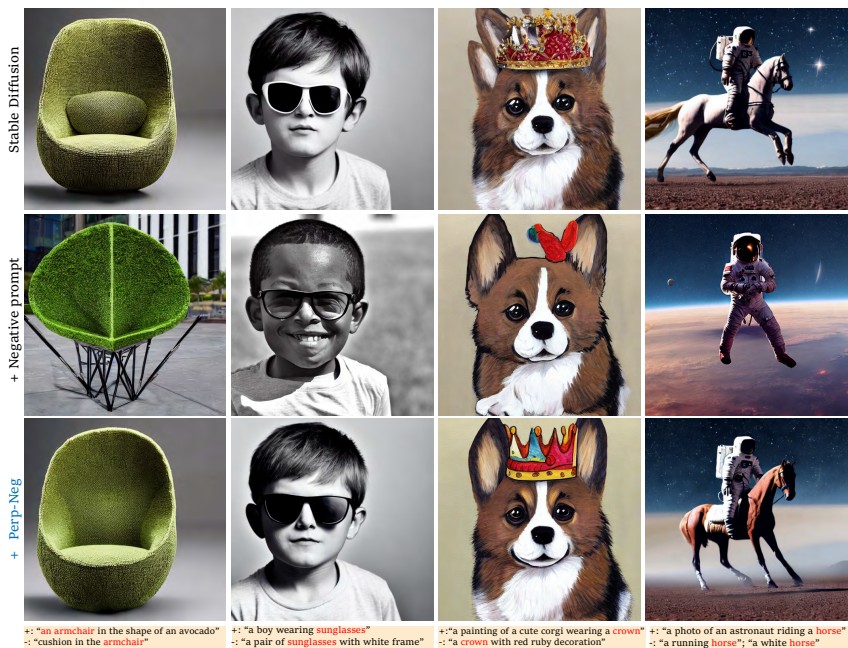

Figure 1: Illustration of Perp-Neg's (training-free) ability to modify generated images using negative prompts while preserving the main concept, for various combinations of positive (+) and negative (-) prompts. *Top to Bottom:* Each column presents the generation from Stable Diffusion (using only positive prompt), Stable Diffusion using both positive and negative prompts, and Stable Diffusion with Perp-Neg sampling. The same seed has been used for the generation of each column.

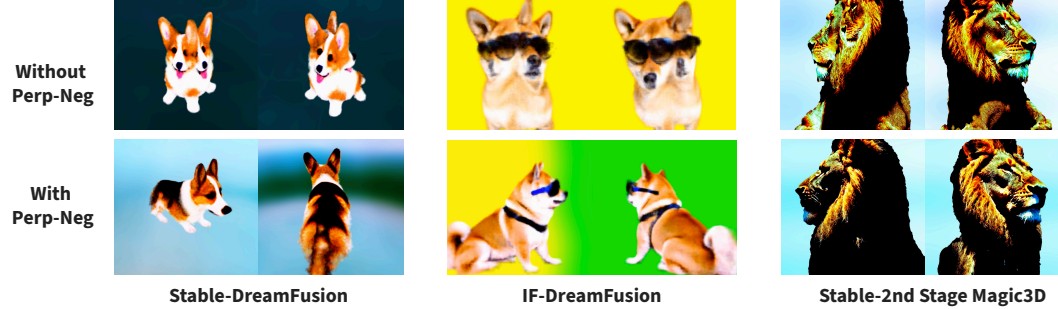

Figure 2: Comparative analysis of 3D generation techniques with and without the 'Perp-Neg' method: left plot - 'DreamFusion with Stable-Diffusion,' middle plot - 'DreamFusion with DeepFloyd-IF,' right plot - 'Magic3D Lin et al. (2022) mesh optimization with Stable-Diffusion.' The results clearly show the effectiveness of 'Perp-Neg' in mitigating the Janus problem in all cases. It is noteworthy that in the leftmost plot, even though we initialized Magic3D with the exact same mesh for both with and without 'Perp-Neg' methods, when 'Perp-Neg' was not used, Magic3D refinement stage resulted in the emergence of two distinct lion faces on the mesh by the end of the training process. The prompts used for generating these assets are "a cute corgi,""a Shiba dog wearing sunglasses," "a DSLR photo of a lion bust."

for the negative prompt. More specifically, Perp-Neg limits the direction of denoising, guided by the negative prompt to be always perpendicular to the direction of the main prompt. In this way, the model is able to eliminate the undesired perspectives in the negative prompts without changing the main semantics, as illustrated in Figure 1.

Furthermore, We extend Perp-Neg to DreamFusion Poole et al. (2022b), a state-of-the-art text-to-3D model, and show how Perp-Neg can alleviate its Janus problem, which refers to the case that a 3D-generated object inaccurately shows the canonical view of the object from several viewpoints, as shown in the top row of Figure 2. Recent studies have considered that the main cause of the Janus problem is the failure of the pre-trained 2D diffusion model in following the view instruction

provided in the prompt Metzer et al. (2022). Therefore, we first, in 2D, show quantitatively and qualitatively how our algorithm can significantly improve the view fidelity of a pretrained diffusion model. Then we integrate Perp-Neg in DreamFusion and Magic3D and show how it can alleviate the Janus problem. Our contributions can be summarized as follows:

- We find the limitations of the current negative prompt implementation which is susceptible to the overlap between a positive and a negative prompt.
- We propose Perp-Neg, a sampling algorithm for text-to-image diffusion models to eliminate undesired attributes indicated by the negative prompt while preserving the main concept, without any training needed.
- Our experiments quantitatively and qualitatively demonstrate that Perp-Neg significantly improves diffusion model prompt fidelity in view generation.
- By enhancing the 2D diffusion model in following the view instruction, we mitigate the Janus problem in text-to-3D generation tasks.

## 2 PERP-NEG: NOVEL NEGATIVE PROMPT ALGORITHM

### 2.1 PRELIMINARY

**Diffusion Models:** Diffusion-based (also known as score-matching) models Ho et al. (2020); Sohl-Dickstein et al. (2015); Song et al. (2021) is a family of generative models that employ a forward process and a reverse process to iteratively corrupt and generate the data within $T$ steps. Specifically, denoting $q(\mathbf{x}_0)$ as the data distribution and $p(\mathbf{x}_T)$ as the generative prior, such two processes can be modeled as the following:

$$\text{forward}: q(\mathbf{x}_{0:T}) = q(\mathbf{x}_0)\prod_{t=1}^{T}q(\mathbf{x}_t|\mathbf{x}_{t-1}), \qquad \text{reverse}: p_\theta(\mathbf{x}_{0:T}) = p(\mathbf{x}_T)\prod_{t=1}^{T}p_\theta(\mathbf{x}_{t-1}|\mathbf{x}_t).$$

One of the most appealing attributes of diffusion models is that any intermediate step of the forward process and every single step in the reverse process can be modeled as a Gaussian distribution like formulated in Ho et al. (2020):

$$q(\mathbf{x}_t|\mathbf{x}_0) = \mathcal{N}(\mathbf{x}_t; \sqrt{\alpha_t}\mathbf{x}_0, (1-\alpha_t)\boldsymbol{I}), \quad p_\theta(\mathbf{x}_{t-1}|\mathbf{x}_t) = \mathcal{N}(\mathbf{x}_{t-1}; \mu_\theta(\mathbf{x}_t, t), \sigma_t^2\boldsymbol{I}).$$

where $\{\alpha_t\}_{t=1}^T$ and $\{\sigma_t\}_{t=1}^T$ can be explicitly calculated with a pre-defined variance schedule $\{\beta_t\}_{t=1}^T$. Moreover, the generator $\mu_\theta(\cdot)$ is a linear combination of $\mathbf{x}_t$ and a trainable generator $\epsilon_\theta$ that predicts the noise in $\mathbf{x}_t$, which is usually optimized with a simple weighted noise prediction loss

$$\theta^\star = \arg\min_{\theta} \mathbb{E}_{t,\mathbf{x}_t,\epsilon} \left[ w(t)\|\epsilon_\theta(\mathbf{x}_t, t) - \epsilon\|^2 \right], \tag{1}$$

$$\mathbf{x}_t = \sqrt{\alpha_t}\mathbf{x}_0 + \sqrt{1-\alpha_t}\epsilon; \ \mathbf{x}_0 \sim q(\mathbf{x}_0), \epsilon \sim \mathcal{N}(\mathbf{0}, \boldsymbol{I}) \tag{2}$$

with $w(t)$ as the weight that depends on the timestep $t$ that is uniformly drawn from $\{1, ..., T\}$.

**Text-to-Image Diffusion Models and Composing Diffusion Model:** Recent works have shown the success of leveraging the power of diffusion models, where large-scale models are able to be trained on extremely large text-image paired datasets by modeling with the loss function in Equation 1 (or its variants) Nichol et al. (2021); Ramesh et al. (2022); Rombach et al. (2022); Saharia et al. (2022), with the text prompt $\mathbf{c}$ often encoded with a pre-trained large language model Devlin et al. (2018); Raffel et al. (2020). To generate photo-realistic images given text prompts, the diffusion models can further take advantage of classifier guidance Dhariwal & Nichol (2021) or classifier-free guidance Ho & Salimans (2022) to improve the image quality. Especially, in the context of text-to-image generation, classifier-free guidance is more widely used, which is usually expressed as a linear interpolation between the conditional and unconditional prediction $\hat{\epsilon}_\theta(\mathbf{x}_t, t, \mathbf{c}) = (1+\tau)\epsilon_\theta(\mathbf{x}_t, t, \mathbf{c}) - \tau\epsilon_\theta(\mathbf{x}_t, t)$ at each timestep $t$ with a guidance scale parameter $\tau$.

When the prompt becomes complex, the model may fail to understand some key elements in the query prompt and create undesired images. To handle complex textual information, Liu et al. (2022) proposes composing diffusion models to factorize the text prompts into a set of text prompts, *i.e.*, $\mathbf{c}=\{\mathbf{c}_1, ...\mathbf{c}_n\}$, and model the conditional distribution as

$$p_\theta(\mathbf{x}|\mathbf{c}_1, ..., \mathbf{c}_n) \propto p(\mathbf{x}, \mathbf{c}_1, ..., \mathbf{c}_n) = p_\theta(\mathbf{x})\prod_{i=1}^{n}p_\theta(\mathbf{c}_i|\mathbf{x}). \tag{3}$$

By applying Bayes rule, we have $p(\mathbf{c}_i|\mathbf{x}) \propto \frac{p(\mathbf{x}|\mathbf{c}_i)}{p(\mathbf{x})}$ and

$$p_\theta(\mathbf{x}|\mathbf{c}_1,...,\mathbf{c}_n) \propto p_\theta(\mathbf{x})\prod_{i=1}^n \frac{p_\theta(\mathbf{x}|\mathbf{c}_i)}{p_\theta(\mathbf{x})}. \tag{4}$$

Note that $p_\theta(\mathbf{x}|\mathbf{c}_i)$ and $p_\theta(\mathbf{x})$ respectively correspond to $\epsilon_\theta(\mathbf{x}_t, t, \mathbf{c}_i)$ and $\epsilon_\theta(\mathbf{x}_t, t)$ modeled by the diffusion model. Putting them together yields a composed noise predictor, as shown in Liu et al. (2022):

$$\hat{\epsilon}_\theta(\mathbf{x}_t, t, \mathbf{c}) = \epsilon_\theta(\mathbf{x}_t, t) + \sum_i w_i \left( \epsilon_\theta(\mathbf{x}_t, t, \mathbf{c}_i) - \epsilon_\theta(\mathbf{x}_t, t) \right), \tag{5}$$

With $w_i$ as a scaling temperature parameter to adjust the weight of the concept components. When one concept $\tilde{\mathbf{c}}$ is needed to be removed, it is proposed to plug in the corresponding component $1/p(\mathbf{x}|\tilde{\mathbf{c}})$ to reformulate Equation 4:

$$p_\theta(\mathbf{x}|\text{not } \tilde{\mathbf{c}}, \mathbf{c}_1,...,\mathbf{c}_n) = p_\theta(\mathbf{x})\frac{p_\theta(\mathbf{x})^\beta}{p_\theta(\mathbf{x}|\tilde{\mathbf{c}})^\beta}\prod_{i=1}^n \frac{p_\theta(\mathbf{x}|\mathbf{c}_i)}{p_\theta(\mathbf{x})},$$

and the corresponding sampler becomes

$$\epsilon_\theta^\star(\mathbf{x}_t, t, \mathbf{c}) = \hat{\epsilon}_\theta(\mathbf{x}_t, t, \mathbf{c}) - w_{\text{neg}} \left( \epsilon_\theta(\mathbf{x}_t, t, \tilde{\mathbf{c}}) - \epsilon_\theta(\mathbf{x}_t, t) \right),$$

where $w_{\text{neg}} > 0$ is a weight function depending on $\tau$ and $\beta$, denoting the scale for the concept negation.

## 2.2 PERPENDICULAR GRADIENT SAMPLING

### 2.2.1 THE PROBLEM OF SEMANTIC OVERLAP

Although Liu et al. (2022) proposes to decompose the text condition into a set of positive and negative prompts in order to help the model handle complex textual inputs, the proposed method assumes these conditional prompts are independent of each other, which requires careful design of the prompts or maybe too ideal to realize in practice. For simplicity of presentation, below we present the overlap problem with the case of fusing two prompts, *i.e.*, the main prompt $\mathbf{c}_1$ and an additional prompt $\mathbf{c}_2$. Without loss of generality, this problem can also be generalized to the case where the main prompt is combined with a series of prompts as $\{\mathbf{c}_1, ..., \mathbf{c}_n\}$. To illustrate the problem, we first re-write the relation in Equation 3:

$$p_\theta(\mathbf{x}, \mathbf{c}_1, \mathbf{c}_2) = p_\theta(\mathbf{x})p_\theta(\mathbf{c}_1|\mathbf{x})p_\theta(\mathbf{c}_2|\mathbf{x})\frac{p_\theta(\mathbf{c}_1, \mathbf{c}_2|\mathbf{x})}{p_\theta(\mathbf{c}_1|\mathbf{x})p_\theta(\mathbf{c}_2|\mathbf{x})}.$$

When $\mathbf{c}_1$ and $\mathbf{c}_2$ are conditional independent given $\mathbf{x}$, the ratio $\mathcal{R}(\mathbf{c}_1, \mathbf{c}_2) = \frac{p_\theta(\mathbf{c}_1, \mathbf{c}_2|\mathbf{x})}{p_\theta(\mathbf{c}_1|\mathbf{x})p_\theta(\mathbf{c}_2|\mathbf{x})} = 1$ and this term can be ignored. However, in practice, the input text prompts can barely be independent when we need to specify the desired attributes of the image, such as style, content, and their relations. When $\mathbf{c}_1$ and $\mathbf{c}_2$ have an overlap in their semantics, simply fusing the concepts could be harmful and result in undesired results, especially in the case of concept negation, as shown in Figure 1. In the second row of images, we can clearly observe the key concepts requested in the main text prompt (respectively "armchair", "sunglasses", "crown", and "horse") are removed when those concepts appear in the negative prompts. This important observation motivates us to rethink the concept composing process and propose the use of a perpendicular gradient in the sampling, which is described in the following section.

### 2.2.2 PERPENDICULAR GRADIENT

Recall when $\mathbf{c}_1$ and $\mathbf{c}_2$ are independent, both of them possess a denoising score component

$$\epsilon_\theta^i = \epsilon_\theta(\mathbf{x}_t, t, \mathbf{c}_i) - \epsilon_\theta(\mathbf{x}_t, t); \ \ i = 1, 2$$

and we can directly fuse these denoising scores as done in Equation 5. However, from the above section, when $\mathbf{c}_1$ and $\mathbf{c}_2$ overlap, we cannot directly fuse the denoising components together, which motivates us to seek the independent component of $\mathbf{c}_2$ to ensure the fused denoising score does not hurt the semantics in $\mathbf{c}_1$.

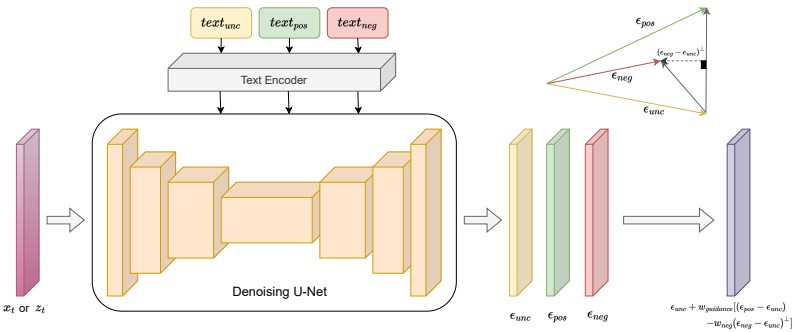

Figure 3: **Overview of Perp-Neg.** The plot shows a denoising step in the Perp-Neg algorithm for the whole scheme of 2D generation, refer to Figure 8 in Appendix

Considering the geometrical interpretation of $\epsilon_\theta^i$ indicates the gradient that the generative model should denoise to produce the final images, a natural solution is to find the perpendicular gradient of $\epsilon_\theta^1$ as the independent component of $\epsilon_\theta^2$. Therefore, we now re-formulate Equation 5 and define the Perp-Neg sampler for $\mathbf{c}_1$ and $\mathbf{c}_2$ as

$$\hat{\epsilon}_\theta^{\text{Perp}}(\mathbf{x}_t, t, \mathbf{c}) = \epsilon_\theta(\mathbf{x}_t, t) + w_1 \epsilon_\theta^1 + w_2 \underbrace{\left( \epsilon_\theta^2 - \frac{\langle \epsilon_\theta^1, \epsilon_\theta^2 \rangle}{\| \epsilon_\theta^1 \|^2} \epsilon_\theta^1 \right)}_{\text{perpendicular gradient}}. \tag{6}$$

where $\langle , \rangle$ denotes the vectorial inner product, $w_1$ and $w_2$ define the weights for each component, and $\frac{\langle \epsilon_\theta^1, \epsilon_\theta^2 \rangle}{\| \epsilon_\theta^1 \|^2}$ defines the projection function to find the most correlated component of $\mathbf{c}_2$ to $\mathbf{c}_1$.

Note that although the proposed perpendicular gradient sampler is applicable for both positive text prompts and negative prompts, we find in the case of concept conjunction, the positive prompts can be designed to be independent of the main prompt in an easier way, as we are creating new details in complementary to the main concept. However, in the case of concept negation, it is more frequent to observe the negative prompts have overlap with the main text prompt. Compared to the sampler in Equation 5, the most important property of the perpendicular gradient is that the component of $\epsilon_\theta^1$ won't be affected by the additional prompt. Imagine the case where $\epsilon_\theta^1 = \epsilon_\theta^2$, using Equation 5, the denoising gradient becomes zero if we also set $w_1 = -w_2$, which might fail the generation. However, using perpendicular gradient in Equation 2.2.2 could still preserve the main component $\epsilon_\theta^1$. Below we mainly discuss the case of using perpendicular gradient sampling to handle the negative prompts and introduce Perp-Neg algorithm.

### 2.2.3 PERP-NEG ALGORITHM

The above section discusses the perpendicular gradient between the main prompt and one additional prompt. Here we generalize it to a set of negative text prompts $\{\tilde{\mathbf{c}}_1, ..., \tilde{\mathbf{c}}_m\}$ and present our Perp-Neg algorithm. We first denote $\mathbf{c}_1$ and $\epsilon_\theta^1$ used in the previous section as $\mathbf{c}_{\text{pos}}$ and $\epsilon_\theta^{\text{pos}}$, which indicate the main positive text prompt condition and the corresponding denoising component, respectively. For any negative text prompt in the set $\tilde{\mathbf{c}}_i$, $i = 1, ..., m$, following equation 2.2.2, the Perp-Neg sampler is defined as

$$\epsilon_\theta^{\text{Perp-Neg}}(\mathbf{x}_t, t, \mathbf{c}_{\text{pos}}, \tilde{\mathbf{c}}_i) = \epsilon_\theta(\mathbf{x}_t, t) + w_{\text{pos}} \epsilon_\theta^{\text{pos}} - \sum_i w_i \underbrace{\left( \epsilon_\theta^i - \frac{\langle \epsilon_\theta^{\text{pos}}, \epsilon_\theta^i \rangle}{\| \epsilon_\theta^{\text{pos}} \|^2} \epsilon_\theta^{\text{pos}} \right)}_{\text{perpendicular gradient of } \epsilon^{\text{pos}} \text{ on } \epsilon^i},$$

with $\epsilon_\theta^i = \epsilon_\theta(\mathbf{x}_t, t, \tilde{\mathbf{c}}_i) - \epsilon_\theta(\mathbf{x}_t, t)$, $w_{\text{pos}} > 0$ and $w_i > 0$ as the weight for positive and each negative prompt. The illustration of Perp-Neg algorithm is shown in Figure 3, and the detailed algorithm is described in Algorithm 1 in Appendix.

## 3    2D DIFFUSION MODEL FOR 3D GENERATION

**Background:** Since 2D diffusion models not only provide samples of density but also allow calculating the derivate of data density likelihood. There are several seminal works that use the latter advantage to uplift a pretrained 2D diffusion and make it a 3D generative model. The main idea behind all these methods is to optimize a 3D scene representation of an object (*e.g.*, NeRF Mildenhall et al. (2021), mesh, *etc.*) based on the likelihood that a diffusion model defines its 2D projections. To be more specific, these algorithms consist of 3 main components:

- 1- A 3D parametrization of the scene $\phi$.
- 2- A differentiable renderer $g$ to create an image $\mathbf{x}$ (or its encoded feature) from a desired camera viewpoint $v$ such that $\mathbf{x} = g(\phi, v)$.
- 3- A pre-trained 2D diffusion model $\theta$ to obtain a proxy of $\log p(\mathbf{x}|\mathbf{c}, v)$ where $p$ is the 2D data density and $\mathbf{c}$ is the text prompt.

The 3D generation has been done as solving an optimization problem as follows:

$$\phi^* = \arg\min_{\phi} \mathbb{E}_v \left[ \mathcal{L}(\mathbf{x} = g(\phi, v)|\mathbf{c}, v; \theta) \right]$$

where $\mathcal{L}$ is a proxy to the negative log-likelihood of the 2D image based on the pre-trained diffusion model.

The noise prediction loss in Equation 1 is a natural choice for $\mathcal{L}$ as the training objective of the diffusion model, since it is a (weighted) evidence lower bound (ELBO) of the data density Ho et al. (2020); Kingma et al. (2021); Poole et al. (2022b):

$$\mathcal{L}_{\text{Diff}} = \mathbb{E}_{t,\epsilon} \left[ w(t) \|\epsilon_\theta(\mathbf{x}_t; t) - \epsilon\|_2^2 \right] \tag{7}$$

However, direct optimization of $\mathcal{L}_{\text{Diff}}$ does not provide realistic samples Poole et al. (2022b). Therefore, Score Distillation Sampling (SDS) has been proposed as a modified version of the diffusion loss gradient $\nabla_\phi \mathcal{L}_{\text{Diff}}$, which is more robust and more computationally efficient as follows:

$$\nabla_\phi \mathcal{L}_{\text{SDS}}(\mathbf{x} = g(\phi)) \triangleq \mathbb{E}_{t,\epsilon} \left[ w(t) \left( \hat{\epsilon}_\theta(\mathbf{x}_t; \mathbf{c}, v, t) - \epsilon \right) \frac{\partial \mathbf{x}}{\partial \phi} \right] \tag{8}$$

where also $\epsilon_\theta$ has been replaced with $\hat{\epsilon}_\theta$ to allow text conditioning by using the classifier-free guidance Ho & Salimans (2022).

Intuitively, this loss perturbs $\mathbf{x}$ with a random amount of noise corresponding to the timestep $t$, and estimates an update direction that follows the score function of the diffusion model to move to a higher-density region.

For the choice of $\mathcal{L}$, since the introduction of the seminal work DreamFusion Poole et al. (2022b), there have been several proposals Lin et al. (2022); Metzer et al. (2022); Wang et al. (2022a). However, since they are similar in core and our method can be applied to all of them, we continue the formulation of the paper by using the Score Distillation Sampling loss presented by DreamFusion.

### 3.1    THE JANUS PROBLEM

Since the introduction of 2D diffusion-based 3D generative models, it has been known that they suffer from the Janus (multi-faced) problem Metzer et al. (2022); Poole et al. (2022b). This refers to a phenomenon that the learned 3D scene, instead of presenting the 3D desired output, shows multiple canonical views of an object in different directions. For instance, when the model is asked to generate a 3D sample of a person/animal, the generated object model has multiple faces of the person/animal (which is their canonical view) instead of having their back view.

View-dependent prompting (*e.g.*, adding back view, side view, or overhead view with respect to the camera position to the main prompt) has been proposed as a remedy but does not fully solve the problem Poole et al. (2022a). We believe part of the reason is that 2D Diffusion models fail to be fully conditioned on the view provided by the prompt, as also pointed out by others Metzer et al. (2022). For instance, when the model is asked to generate the back view of a peacock, it wrongly

produces the front view instead, as the front view has been more prominent in the training data the model has been trained on.

To provide an intuitive mathematical understanding of the Janus problem, we believe one of the reasons is the model fails to be properly conditioned on view $v$. More specifically, the proxy of $\log p(\mathbf{x}|\mathbf{c}, v)$ does not fully restrict $\mathbf{x}$ to have zero density on areas that do not represent the viewpoint $v$ for the scene description $y$. The main reason we think this is the case is samples of the density fail to reflect the direction of interest.

## 3.2 PERP-NEG TO ALLEVIATE JANUS PROBLEM AND 2D VIEW CONDITIONING

In this section, we first explain how combining Perp-Neg with a unique prompting technique can enable us to accurately condition the 2D diffusion model on the desired view. Additionally, we will explore how Perp-Neg can be integrated with SDS loss to address the Janus problem by improving the view faithfulness of the 2D model.

To begin, we demonstrate how to generate a desired statistical view using the improved model. Then, we explain the process for creating interpolations between two views To generate a specific view of an object, we use a combination of positive and negative prompts. We define $\textbf{txt}_{back}$, $\textbf{txt}_{side}$, and $\textbf{txt}_{front}$ as the main text prompts appended by back, side, and front views, respectively. We replace simple prompts containing the view with the following set of positive and negative prompts to generate each view:

$$\textbf{txt}_{back} \to [+\textbf{txt}_{back}, -w^{\text{b}}_{side}\textbf{txt}_{side}, -w^{\text{b}}_{front}\textbf{txt}_{front}]$$

$$\textbf{txt}_{side} \to [+\textbf{txt}_{side}, -w^{\text{s}}_{front}\textbf{txt}_{front}], \qquad \textbf{txt}_{front} \to [+\textbf{txt}_{front}, -w^{\text{f}}_{side}\textbf{txt}_{side}]$$

where $w_{(\cdot)} \geq 0$ denotes the weights for the negative prompts. Positive and negative prompts are fed into the Perp-Neg algorithm during each iteration of the diffusion model. We don't include $\textbf{txt}_{back}$ as a negative prompt for the generation of side/front views since most objects' canonical view is not back. However, if the back view is more prominent for some objects, it should be included as a negative prompt. We also observed increasing the weight of the negative prompt makes the algorithm focus more on avoiding that view, acting as a pose factor.

In this subsection, we will first explain how we interpolate between the side and back views, followed by the interpolation between the front and side views. We distinguish between these two cases because the diffusion model may be biased toward generating front views, and if this assumption is not true, then the formulation needs to be adjusted accordingly.

To interpolate between the side and back views, we use the following embedding as the positive prompt:

$$r_{inter} * \textbf{emb}_{side} + (1 - r_{inter})\textbf{emb}_{back}; \qquad 0 \leq r_{inter} \leq 1$$

where $\textbf{emb}_v$ is the encoded text for the view $v$ and $r_{inter}$ is the degree of interpolation. And for the negative prompts, we use:

$$[-f_{\text{sb}}(r_{inter})\textbf{txt}_{side}, -f_{\text{fsb}}(r_{inter})\textbf{txt}_{front}]$$

such that $f_{\text{sb}}, f_{\text{fsb}}$ are positive decreasing functions. The second negative prompt is chosen based on the assumption that the diffusion model is more biased towards generating samples from the front view.

For interpolation between the front and side views, the embedding for the positive would be:

$$r_{inter} * \textbf{emb}_{front} + (1 - r_{inter})\textbf{emb}_{side}$$

and the following two negative prompts

$$[-f_{\text{fs}}(r_{inter})\textbf{txt}_{front}, -f_{\text{sf}}(1 - r_{inter})\textbf{txt}_{side}]$$

where $f_{\text{fs}}(1), f_{\text{sf}}(1) \approx 0$ and both of the functions are decreasing.

**Perp-Neg SDS:** We employed interpolation technique in Stable DreamFusion and varied $r_{inter}$ based on the related direction of 3D to 2D rendering. To be more specific, we modified the SDS loss [8] as follows:

$$\nabla_\phi \mathcal{L}^{PN}_{\text{SDS}} \triangleq \mathbb{E}_{t,\epsilon} \left[ w(t) \left( \hat{\epsilon}^{PN}_\theta(\mathbf{x}_t; \mathbf{c}, v, t) - \epsilon \right) \frac{\partial \mathbf{x}}{\partial \phi} \right] \tag{9}$$

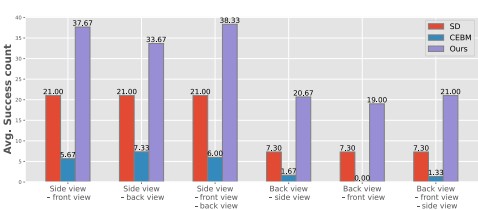

Figure 4: Averaged successful generation count in terms of different positive and negative prompt combinations.

| Method | Side view | Back view |
|---|---|---|
| Stable Diffusion | 42.0% | 14.6% |
| CEBM | 12.7% | 2.0% |
| Perp-Neg (Ours) | **73.1**% | **40.4**% |

Figure 5: Comparison of successful generation rate.

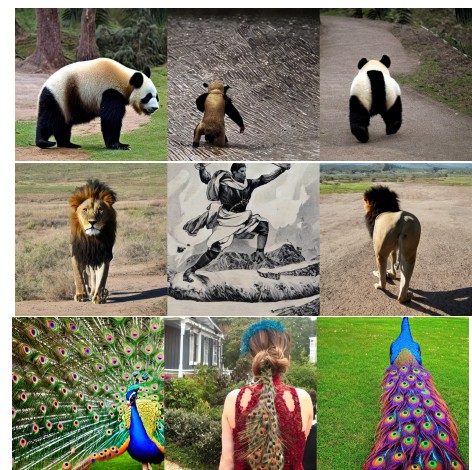

Figure 6: Comparison of generation of the back view of panda, lion and peacock using the vanilla sampler, CEBM, and our Perp-Neg (*from left to right*) with Stable Diffusion.

such that $\hat{\epsilon}_\theta^{PN}(\mathbf{x}_t; \mathbf{c}, v, t)$ is:

$$\epsilon_\theta^{\text{unc}} + w_{guidance}\big[\epsilon_\theta^{\text{pos}_v} - \sum_i w_v^{(i)}\epsilon_\theta^{\text{neg}(i)\perp}\big]. \tag{10}$$

The unconditional term $\epsilon_\theta^{unc}$ refers to $\epsilon_\theta(\mathbf{x}_t, t)$, and

$$\epsilon_\theta^{\text{pos}_v} = \epsilon_\theta(\mathbf{x}_t, t, c_{\text{pos}}^{(v)}) - \epsilon_\theta(\mathbf{x}_t, t), \qquad \epsilon_\theta^{\text{neg}(i)} = \epsilon_\theta(\mathbf{x}_t, t, c_{\text{neg}(i)}^{(v)}) - \epsilon_\theta(\mathbf{x}_t, t)$$

where $c^{(v)}$ refers to the text embedding of positive/negative at direction $v$. And $\epsilon_\theta^{\text{neg}(i)\perp}$ is the perpendical component of $\epsilon_\theta^{\text{neg}(i)}$ on $\epsilon_\theta^{\text{pos}_v}$. And, $w_v$'s are representative of the weights of the negative prompts at direction $v$.

## 4 EXPERIMENTS

In this section, we first conduct experiments on 2D cases to quantitatively demonstrate the importance of using Perp-Neg in the sampling to improve the likelihood of getting the image corresponding to the text query, which provides evidence of why our method surpasses vanilla sampling in the 3D case. Next, we show results in 3D generation.

### 4.1 STATISTICS ON SEMANTIC-ALIGNED 2D GENERATIONS

In the first experiment, we set the random seeds as 0-49 to get 50 images from each text prompt. We carefully select qualified images that align with the requested text based on a series of criteria and report the percentage of accepted samples produced with Stable Diffusion, Compositional Energy-based Model (CEBM), and our Perp-Neg. Below we introduce the details of prompt design and the criteria for accepting qualified samples.

**Design of prompts:** We design the basic text prompts as: "A [O], [V] view." Token [O] stands for the objects, such as panda, lion; token [V] stands for view, where we only consider "front", "back" and "side" in our experiments. For example, we use "A panda, side view" to request the model to generate an image showing the side view of a panda. For more detail please refer to Appendix A.2.

**Average success rate:** We test each group of prompts using three objects, "panda", "lion", and "peacock" and only count photo-realistic generation that matches the text prompt query as a successful generation. For detailed acceptance criteria and other details, please refer to Appendix A.2. We count the averaged percentage of accepted successful generations, summarized in Table 5.

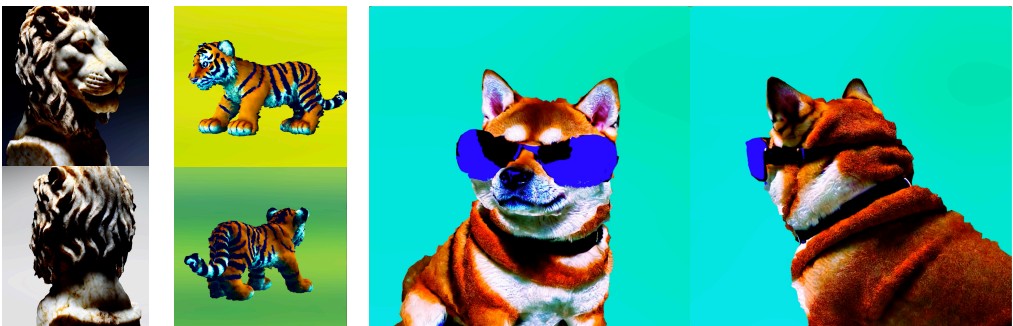

Figure 7: Qualitative examples of Magic3D with Perp-Neg. The prompts used for generating these assets are "a tiger cub," "a marble lion bust," "a Shiba dog wearing sunglasses." **It is important to highlight that, in the absence of Perp-Neg, the Magic3D model, whether utilizing Stable-Diffusion or DeepFloyd-IF, failed to produce satisfactory results across all the above prompts, even after six attempts.**

As shown, we can observe the vanilla sampling from Stable Diffusion only has 42.0% in successfully generating requested side-view images. For more difficult cases like generating the back-view images, the success rate is even lowered to 14.6%. By simply using negative prompts without considering the overlap between positive and negative prompts, CEBM fails to generate the desired view and results in a lower success rate compared to Stable Diffusion. Figure 6 shows a qualitative justification corresponding to Table 5.

**On the combination of positive and negative prompts:** To explore how to combine negative prompts with the positive prompt. We compute the averaged successful generation count across all tested objects and report the averaged count using different positive and negative prompt combinations in Figure 4. From the figure, it is notable that when generating the side-view images, using the back view as the negative prompt is less effective than using the front view or using the combination of both the front view and back view. Similarly, when generating back-view images, using the front view in negative prompts is also less effective, since the model is less likely to generate front-view details when conditioned on the back view, while the side view is more ambiguous to the model.

## 4.2 PERP-NEG FOR 3D

To demonstrate the efficacy of Perp-Neg in mitigating the Janus problem across various scenarios, we employ Perp-Neg with two distinct choices of pre-trained diffusion models: Stable-Diffusion and DeepFloyd-IF. We take this approach to underscore that our algorithm's utility extends beyond specific models or architectures.

Furthermore, we seek to illustrate that Perp-Neg can offer value beyond its application in the Dream-Fusion algorithm. To this end, we integrate it into the Magic3D framework, showcasing its effectiveness in refining 3D objects without causing the Janus problem. A selection of our qualitative observations can be found in Figures 2 and 7. For a more comprehensive exploration and details, including quantitative comparisons, please refer to Appendix B.

## 5 CONCLUSION

We introduce Perp-Neg, a new algorithm that enables negative prompts to overlap with positive prompts without damaging the main concept. Perp-Neg provides greater flexibility in generating images by enabling users to edit out unwanted concepts from initial generated photos. More importantly, Perp-Neg enhances prompt faithfulness by preventing the 2D diffusion model from producing biased samples from its training data and accurately representing the input prompt. This can be accomplished by feeding to Perp-Neg a sentence describing the model bias as the negative prompt to generate desired solutions. Our paper also demonstrates how Perp-Neg can properly condition the 2D diffusion model to generate views of interest rather than a canonical view. Finally, we integrate Perp-Neg's robust view conditioning property into SDS-based text to 3D models and show how it alleviates the Janus problem.

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
