# OpenReview forum: "Re-imagine the Negative Prompt Algorithm for 2D/3D Diffusion"
_ICLR.cc/2024/Conference — Submitted to ICLR 2024_

### Official Review · Reviewer_6H1W · 2023-10-16

**Soundness:** 4 excellent
**Presentation:** 4 excellent
**Contribution:** 4 excellent
**Rating:** 6
**Confidence:** 4

**Summary:**

This paper solves the training data bias problem in the text-to-image model using a new formulation of negative prompt: Perp-Neg. This paper first presents the perp-neg algorithm where the latents are updated in the perpendicular gradient of “negative prompt- unconditional prompt“ with “positive prompt - unconditional prompt”. Then, this idea is utilized in solving 2D image generation with Janus problem in text-to-3d.

**Strengths:**

It is clever to solve Janus problem using negative prompts.
The illustration of the algorithm is very easy to understand and plausible.
Code is provided to reproduce the results.

**Weaknesses:**

The visual results in the paper are good and interesting.
A quantitative successful generation rate is also provided. My only concern is result part lacks quantitative fidelity results like FID, clip similarity, and user preference.

**Questions:**

I am happy to see more quantitative results

---

> ### Author Response · Authors · 2023-11-22
> **Response to Reviewer 6H1W**
>
> Dear Reviewer 6H1W,
>
> Thank you for your thorough review and positive feedback on our paper. We appreciate your recognition of our novel approach in solving the Janus problem using a novel negative prompt algorithm.
>
> We acknowledge your suggestion regarding the inclusion of quantitative fidelity results like FID, CLIP similarity, and user preference. We agree that these metrics will enhance our paper's robustness and will include them in our revised submission.
>
> Thank you again for your support and positive comments on the value of our work!!

---

### Official Review · Reviewer_j2ct · 2023-10-29

**Soundness:** 2 fair
**Presentation:** 2 fair
**Contribution:** 3 good
**Rating:** 6
**Confidence:** 4

**Summary:**

The paper proposes Perp-Neg, a method of using negative prompting without the negative effect of semantic overlap. The basic idea is to find the component of the negative prompt gradient that is orthogonal to the original prompt gradient. The authors show that this method can be used as a more effective negative prompting method, and also helps alleviate the Janus problem in score distillation for 3D generation.

**Strengths:**

1. The motivation (Section 2.2.1) is insightful. Semantic overlap seems to be an important drawback of naive negative prompting, and is not mentioned in previous papers before as far as I know.

2. The proposed method (Section 2.2.2 and 2.2.3) is elegant and intuitive.

3. The proposed method shows good qualitative results in generating view-dependent images (Figure 6).

**Weaknesses:**

1. The proposed method seems to be applicable in many tasks where gradients are mixed during inference (See [1] for an example). But the authors only focus on alleviating the Janus problem in 3D generation. I think it would be a stronger paper if more applications are included.

2. The number of qualitative results of 3D generation is too small. All I can find is Figure 2, 7, 20, which contain 9 examples with overlapping prompts (e.g. lions). The Janus problem is very common in score distillation based generation methods, so more such comparisons with a diverse set of prompts should be presented.

3. The qualitative results for view-dependent image generation (Figure 6, 10-18) re-use the same prompts (lion, panda, peacock) over and over again. It is better to use more different prompts to showcase the effectiveness of the method in the wild.

4. The comparison to other methods of 3D generation is limited and incomplete. I only see it in Figure 2 with 3 examples, and the last one for Magic3D does not show obvious difference to me. Weirdly in Figure 7, the captions claims the original Magic3D model fails to generate satisfactory results, but the figures do not show any of these "unsatisfactory" results.

[1] Universal Guidance for Diffusion Models, Bansal et al.

**Questions:**

I think this method is interesting, but why do you focus so much on the Janus problem of 3D generation, instead of trying to demonstrate the method's effectiveness on more applications?

---

> ### Author Response · Authors · 2023-11-22
> **Response to Reviewer j2ct**
>
> Dear Reviewer j2ct,
>
> Thank you for your thorough review and insightful observations regarding our submission. Your feedback has been invaluable in enhancing the quality and scope of our research.
>
> **1-Expanding the Scope Beyond 3D Generation:**
>
> We appreciate your emphasis on the broader potential applications of our Perp-Neg methodology. Initially, our focus was predominantly on 3D generation due to its profound impact and the specific challenges presented by the Janus problem in score distillation-based methods. Recognizing the value of your suggestion, we will extend our research to encompass additional applications, such as segmentation mapping and guided bounding box generation, and leave the rest for future work.
>
> **2-More 3D Experiment**
>
>  We thank you for pointing out the need for a wider array of examples. Initially, we presented results from 12 prompts, totaling 240 experiments. Based on your valuable feedback, we expanded our dataset to include 10 additional prompts, each tested with and without Perp-Neg. The diversity and increased number of experiments strengthen our findings, as detailed in the revised Table.
>
> Please refer to the response provided to reviewer vLLW question 2nd.
>
> **3- More 2D view Experiment**
>
> For view-dependent images, we've now included varied and creative prompts, enhancing the demonstration of our method's effectiveness in 'real-world' scenarios. The updated results are as follows:
>
> | Prompt                                                 | SD Side View (%) | Ours Side View (%) | SD Back View (%) | Ours Back View (%) |
> |--------------------------------------------------------|------------------|--------------------|------------------|--------------------|
> | Michelangelo style statue of an astronaut              | 37.6             | 62.3               | 22.9             | 47.6               |
> | A blue poison-dart frog sitting on a water lily        | 46.9             | 71.7               | 32.1             | 56.9               |
> | A humanoid robot using a laptop                        | 47.9             | 68.9               | 36.9             | 57.9               |
> | A kangaroo sitting on a bench playing the accordion    | 39.6             | 60.1               | 22.2             | 42.7               |
> | A rabbit cutting grass with a lawnmower                | 42.6             | 66.3               | 28.9             | 52.6               |
> | A teddy bear pushing a shopping cart full of fruits... | 51.2             | 74.0               | 38.4             | 61.2               |
> | A high quality photo of a classic silver muscle car    | 78.4             | 93.2               | 64.6             | 88.4               |
> | A DSLR photo of a yellow duck                          | 46.8             | 70.7               | 16.4             | 28.4               |
> | A photo of a horse walking                             | 46.3             | 78.5               | 36.1             | 68.2               |
> | A chimpanzee dressed as a football player              | 39.9             | 64.8               | 25.0             | 49.9               |
>
>
> **4(a)- More 3D Baseline**
>
> Thanks for raising this point! Our intention was not to directly compete with existing text-to-3D models but to show how Perp-Neg can enhance them as a plug-in when combined with them. We will clarify this in the paper. The provided examples, highlight the significant reduction of the Janus problem when our method is integrated into these baselines.
>
> **4(b)- Figure 7 Clarification**
>
> To clarify, Figure 7 in our paper is intended to demonstrate the improved results achieved by integrating Perp-Neg with the Magic3D model. We acknowledge that the absence of standalone results from "Magic3D without Perp-Neg" might have caused a lack of context and confusion.
>
> Figure 7 only shows the result of "Magic3D + Perp-Neg." To make things clearer, we'll add some pictures to the appendix of our paper. These pictures will show what happens when we use "Magic3D without Perp-Neg" method. This will help show why our method is important and how it fixes the problems with Magic3D. For all tries "Magic3D without Perp-Neg" we observed the Janus problem.
>
> We are truly grateful for your comprehensive review and constructive feedback, which have been instrumental in enhancing the quality and clarity of our paper. Thank you so much!

---

> > ### Comment · Reviewer_j2ct · 2023-11-23
> >
> > Thanks for the additional experiments. I will increase my rating to 6.

---

### Official Review · Reviewer_Vwgv · 2023-11-01

**Soundness:** 3 good
**Presentation:** 3 good
**Contribution:** 2 fair
**Rating:** 5
**Confidence:** 5

**Summary:**

The submission discusses the limitations of text-to-image diffusion models that the 2D results of diffusion models can not align exactly with the provided prompt in terms of the view angles. To address this issue, the authors propose a new algorithm called Perp-Neg, which leverages geometrical properties of the score space and does not require any training or fine-tuning of the model. The algorithm can be applied to both 2D and 3D generation. The paper shows the effectiveness of Perp-Neg in addressing the Janus problem in 3D object generation, the results show a fair improvement in the success rate of side/back views, leading to more view-consistent 3D objects. Additionally, the appendix contains sufficient ablations that confirm the importance of the design choices.

**Strengths:**

While the proposed method is simple, the authors have demonstrated its efficacy in 2D/3D diffusion. In addition, it does not require offline training and fine-tuning of the image diffusion model and can preserve the generalizability of the original diffusion model to a great extent.

The paper is clearly written and well organized, and the pipeline described in this submission is technically sound and reproducible.

Sufficient evaluations, such as the quantitative results of the success rate and qualitative results of the generated images of the side and back sides.

**Weaknesses:**

One of the most important applications of the proposed method is to boost the text-to-3D generation task. However, the experiment results shown in the paper mainly focus on 2D images on the side/back views, rather than the 3D generation.

The generated 3D objects are rather simple, so the effectiveness of the proposed method in the 3D generation of detailed objects needs to be further verified.

The comparison is a bit weak, it seems like CEBM is not designed for the task of this submission. Instead, the work [1], which also focuses on the view-consistent text-to-3D generation, should be cited and compared.

[1] Hong et. al., Debiasing Scores and Prompts of 2D Diffusion for View-consistent Text-to-3D Generation, NIPS 2023

**Questions:**

What is the Compositional Energy-based Model (CEBM) and why choose it as a competing baseline? The paper does not provide an explanation for this choice, nor does it refer to any related work for CEBM.

What are the consequences of using just negative prompts instead of the proposed Perp-Neg in 3D generation? I think simply using negative prompts the regulate the 2D results on the back/side views may also alleviate the janus problem.

---

> ### Author Response · Authors · 2023-11-22
> **Response to Reviewer Vwgv**
>
> Dear Reviewer Vwgv,
>
> We are truly grateful for your valuable feedback on our manuscript. We have carefully considered your comments and have made several enhancements to our paper to address your concerns.
>
> **1-Focus on 2D Images and Relevance to 3D Generation:**
>
> Thanks for raising this point! We will clarify in the paper that in our initial submission, we conducted 240 trials of text-to-3D generation, incorporating 12 different prompts. Our emphasis on 2D was primarily due to the significant computational demands of 3D experiments. Each 3D trial required over 30 minutes of processing, compared to just 8 seconds for 2D.
>
> Additionally, we found that improvements in 2D were closely linked to enhancements in 3D generation, offering a more efficient pathway to examine the root cause of the Janus problem. We will add this clarification to the revised manuscript to better explain our methodology and focus.
>
> **2-Extended Experiments and Comparative Analysis with Paper [1]:**
>
> In response to your suggestion, we expanded our experimental scope. We conducted an additional 180 experiments for each prompt, with variations including the use of Perp-Neg, no Perp-Neg, and prompt debiasing (using DeepFloyd-IF). These experiments have provided more comprehensive insights into the effectiveness of our proposed method, especially in comparison to the work presented in Hong et al. (2023). We will include these additional results and a more thorough comparative analysis in our revised manuscript.
>
> Here are the updated results from these experiments:
>
> | Prompt                                                         | % Success with Perp-Neg | % Success without Perp-Neg | % Success with Prompt Debiassing |
> |----------------------------------------------------------------|-------------------------|----------------------------|----------------------------------|
> | Michelangelo style statue of an astronaut                      | 50.00%                  | 16.67%                     | 33.33%                           |
> | Blue poison-dart frog sitting on a water lily                  | 66.67%                  | 33.33%                     | 66.67%                           |
> | Humanoid robot using a laptop                                  | 50.00%                  | 16.67%                     | 33.33%                           |
> | Kangaroo sitting on a bench playing the accordion              | 50.00%                  | 0.00%                      | 33.33%                           |
> | Rabbit cutting grass with a lawnmower                          | 66.67%                  | 33.33%                     | 50.00%                           |
> | Teddy bear pushing a shopping cart full of fruits and vegetables | 50.00%               | 0.00%                      | 16.67%                           |
> | High quality photo of a classic silver muscle car              | 83.33%                  | 83.33%                     | 83.33%                           |
> | DSLR photo of a yellow duck                                    | 33.33%                  | 0.00%                      | 33.33%                           |
> | Photo of a horse walking                                       | 33.33%                  | 0.00%                      | 33.33%                           |
> | Chimpanzee dressed as a football player                        | 33.33%                  | 16.67%                     | 16.67%                           |
>
>
> These findings reinforce our initial conclusions and demonstrate the practical effectiveness of Perp-Neg in various scenarios.
>
> **3-Simplicity of Generated 3D Objects and Detailed Object Generation:**
>
> Thank you so much for this comment! Our choice of simple prompts was intentional, as we aimed to isolate and address the view consistency issue without the confounding factor of complex object generation. By confounding factors, we refer to the problem of the text-to-image model missing some objects provided in the prompt. However, following your suggestion, we have now also tested our algorithm with more complex prompts and provided the result above.
>
> **4-Choice of CEBM as a Baseline and Exploration of Negative Prompts:**
>
> Thanks for this question! We would like to clarify that the Compositional Energy-based Model (CEBM) is exactly the naive negative prompt algorithm. CBEM just mathematically motivates why the negative prompt algorithm works.  In the revised version of our paper, we will clarify this distinction more explicitly to ensure a better understanding of why CEBM was referenced
>
>  We want to express our sincere gratitude for the time and effort you've dedicated to reviewing our paper.  Thank you again for your invaluable contribution to our work.

---

### Official Review · Reviewer_vLLW · 2023-11-03

**Soundness:** 3 good
**Presentation:** 3 good
**Contribution:** 2 fair
**Rating:** 5
**Confidence:** 5

**Summary:**

The paper proposed a Perp-Neg method to control the text-to-image diffusion model by pependicular gradient sampling. This method leverage the geometry properties of the score space in diffusion models.  Authors present experiments of this method in 2d image translation and 3D generative models with SDS to mitigate the janus problem in text-to-3D task.

**Strengths:**

1. The proposed pependicular gradient sampling is performed on the latent noise space of diffusion models, is easy to follow.
2. This method gives a tractable method to balance betweet the score of postive and negative text prompts.
3. The method is easy to implement and follow.

**Weaknesses:**

1. The assumption of this method is based on the pependicular gradient sampling, but this design is somewhat heuristic, and have not been proved in the paper.
2. Weakness 1 caused the effectiveness of this method on text-to-3D is limited actually. Janus problems almost cannot be mitigated according to implementation in threestudio. [1]
3. The balance factor defined between eq.8 and eq.9 may be hard to tune in experiments

[1] https://github.com/threestudio-project/threestudio/issues/8

**Questions:**

please refer to the weakness part.

---

> ### Author Response · Authors · 2023-11-22
> **Response to the reviewer vLLW**
>
> Dear Reviewer vLLW,
>
> We greatly appreciate your comprehensive review and constructive feedback on our paper. Your insights are invaluable in refining our paper!
>
> **1-Regarding Perpendicular Gradient Sampling Theory:**
>
> We acknowledge your concern regarding the lack of formal mathematical proof for our perp-neg method. While a formal mathematical proof is not presented, we believe our geometric reasoning and mathematical motivations, detailed in sections 2.2.1 and 2.2.2, provide a solid foundation for our approach. We appreciate your pointing out this limitation and will consider adding a more rigorous theoretical underpinning in future work.
>
> **2-Effectiveness of Perp-Neg in alleviating the Janus Problem - Threestudio:**
>
> We acknowledge your concern about our method's limited effectiveness in text-to-3D as highlighted in the Threestudio implementation.
>
> It is essential to clarify that the Threestudio version is a derivative of our original algorithm and does not mirror our methodology or results. In our experiments, we observed a significant reduction in the Janus problem, as presented in the appendix.
>
> In our experiments, comprising 240 trials (120 with Perp-Neg and 120 without), we observed a significant mitigation of the Janus problem. The following table, which we plan to include in the revised manuscript, details our findings:
>
> | Algorithm                     | Perp-Neg Usage    | Acceptance Rate | Total Trials | Total Prompts |
> |-------------------------------|-------------------|-----------------|--------------|---------------|
> | Stable-Diffusion DreamFusion  | With Perp-Neg     | 20.24%          | 84           | 6             |
> | Stable-Diffusion DreamFusion  | Without Perp-Neg  | 7.14%           | 84           | 6             |
> | DeepFloyd-IF DreamFusion      | With Perp-Neg     | 50.00%          | 36           | 6             |
> | DeepFloyd-IF DreamFusion      | Without Perp-Neg  | 2.78%           | 36           | 6             |
>
> Overall, our method with Perp-Neg demonstrated a 29.17% improvement over the non-Perp-Neg approach (5.83%). These results are presented in the original manuscript with 12 different prompts in total. We have mentioned the success rate for each prompt in the paper.
>
>
> It's crucial to note that while we haven't entirely solved the Janus problem, our method offers a significant advancement in addressing it. We will clarify this in the paper.
>
> Following your suggestion and other reviewers, we will also include an additional 120 experiment results on text-to-3D experiments in the appendix to further substantiate our claims. The table of the new results is presented in the following:
>
> | Prompt                                                             | % Success with Perp-Neg | % Success without Perp-Neg |
> |:-------------------------------------------------------------------|------------------------:|---------------------------:|
> | Michelangelo style statue of an astronaut                          |                 50.00%  |                    16.67% |
> | A blue poison-dart frog sitting on a water lily                    |                 66.67%  |                    33.33% |
> | A humanoid robot using a laptop                                    |                 50.00%  |                    16.67% |
> | A kangaroo sitting on a bench playing the accordion                |                 50.00%  |                     0.00% |
> | A rabbit cutting grass with a lawnmower                            |                 66.67%  |                    33.33% |
> | A teddy bear pushing a shopping cart full of fruits and vegetables |                 50.00%  |                     0.00% |
> | A high quality photo of a classic silver muscle car                |                 83.33%  |                    83.33% |
> | A DSLR photo of a yellow duck                                      |                 33.33%  |                     0.00% |
> | A photo of a horse walking                                         |                 33.33%  |                     0.00% |
> | A chimpanzee dressed as a football player                          |                 33.33%  |                    16.67% |
>
>
> **3-On Tuning the Balance Factor Between Equations 8 and 9:**
>
> We appreciate your concern about the balance factor's tuning difficulty. We wanted to clarify that Equation 9 is used across all text-to-3D tasks. Regarding the weight of negative prompts, we found that the model is not sensitive to the weight of negative prompts if they are beyond a certain threshold, as observed in both 2D and 3D settings. This observation will be elaborated in our revised manuscript for better clarity.
>
> Thank you once again for your thorough evaluation and constructive criticism. Your feedback has been instrumental in refining our work, and we are confident that these revisions will significantly improve our paper.

---

> ### Comment · Reviewer_vLLW · 2023-11-23
> **Response to authors**
>
> Dear authors,
>
> I appreciate your response. These added experiments have addressed some of my concerns.
> However, I still have doubts about the effectiveness of your methods in text-to-3D models.
>
> I have tried your released code, but it is still prone to generating Janus-faced objects.
> The interpolation weight between the negative prompts and positive prompts is hard to decide; it needs to be tuned case by case.
>
> Besides, I also tried your code on 2D-image generation. I found that the generated image may encounter distortions and oversaturation when adopting your proposed method. I speculate it's the perp-neg sampling method in Equation 6 that poses an OOD issue for the latent decoder of diffusion models.
>
> So, I tend to keep my previous rating.

---

### Author Response · Authors · 2023-11-23
**Appreciation for Review and Request for Additional Feedback Prior to Deadline**

Dear Reviewers,

We deeply appreciate your thorough review and valuable feedback on our paper. Your constructive comments have significantly contributed to its improvement. With the deadline nearing, we kindly ask you to inform us of any additional concerns or follow-up questions.

Your insights are crucial, and we are fully prepared to address any further clarifications or issues that may arise.

Thank you again for your dedication and effort in reviewing our work.

---

### Meta-Review · Area_Chair_RUM7 · 2023-12-10

**Metareview:**

The manuscript presents the 'Perp-Neg' method, a novel method to control text-to-image diffusion models through perpendicular gradient sampling. This work has been reviewed by four experts in the field, resulting in mixed reviews. The reviewers have acknowledged the strength of this paper: the proposed method's ability to operate without offline training and fine-tuning, while largely preserving the generalizability of the original diffusion model.

However, after careful consideration of the rebuttal and further deliberation, the reviewers maintain certain reservations. A primary concern is the method's effectiveness in 3D generation, particularly for fine-grained objects. This aspect is crucial for demonstrating the versatility and applicability of the 'Perp-Neg' method. The reviewers suggest that a more comprehensive evaluation in this aspect would greatly enhance the paper's validity.

While the paper certainly introduces a novel approach, the decision at this stage is to not recommend acceptance. We encourage the authors to take the reviewers' feedback into consideration, particularly focusing on substantiating the effectiveness of their method in the 3D generation of fine-grained objects.

**Justification For Why Not Higher Score:**

The effectiveness of the proposed method in the 3D generation of fine-grained objects needs to be further verified.

**Justification For Why Not Lower Score:**

N/A

---

### Decision · Program_Chairs · 2024-01-16

Reject